

# CO₂ Flux and its Relationship with Water Parameters and Biological Activity in the Ji-Paraná River (Rondônia State – Western Amazon)

Thandy Junio da Silva Pinto[1], Beatriz Machado Gomes[2]

[1]Post-Graduate Program in Environmental Engineering Sciences, Center of Water Resources and Environmental Studies, Nucleus of Ecotoxicology and Applied Ecology, University of São Paulo, Rodovia Domingos Innocentini, km 13, 13560-970, Itirapina (SP), Brazil. Post Office Box 292, São Carlos, SP CEP 13560-970, Brasil
[2]Laboratory of Hydrogeochemistry, Environmental Engeneering Departament, Federal University of Rondônia
Rua Rio Amazonas 351, Jardim dos Migrantes, CEP 76900-726

*Correspondence to:* Thandy Junio da Silva Pinto (thandyjuniosilva@usp.br)

**Abstract.** Over the past decades, the role of tropical rivers in the carbon cycle paradigm has been revised due to their importance in the regional carbon budget and identified as important emitters of carbon to atmosphere. This study, conducted in the Ji-Paraná River, quantifies the CO₂ flux in the water-air interface according to the floating chamber methodology and verifies the seasonal variation and its relationship with physical, chemical and biological water

parameters. The physical-chemical parameters analyzed were temperature, pH, electrical conductivity, dissolved oxygen, transparency, alkalinity, and turbidity, and the biological parameter was chlorophyll concentration. The Ji-Paraná River behaves as a source of carbon to the atmosphere, with higher emission peaks in high waters ($4869.20 \pm 779.95$ mgC m$^{-2}$ day$^{-1}$) and lower values in low water ($711.20 \pm 131.51$ mgC m$^{-2}$ day$^{-1}$) - the difference between the periods reached 6.8 times. Dissolved oxygen, pH, electrical conductivity and water transparency were correlated with flux and explained the

gaseous exchange dynamics. CO₂ flux was strongly related to the river flow, which evidences the importance of river regime variations in gaseous emissions. Chlorophyll was strongly related to flux, therefore, the primary production is an important sink of CO₂ to the Ji-Paraná River, which indicates biological activity significantly influences the river carbon cycling.

## 1 Introduction

The increasing effects of greenhouse gases on the atmosphere and the global climate have been extensively discussed and research has been conducted towards the quantification of the sources and sinks of carbon for the atmosphere. Recent studies have focused on the strong influence of rivers on the regional and global carbon balance, since they can behave as sinks or sources of atmospheric carbon (e.g. Richey et al., 2002; Ciais et al., 2008; Johnson et al., 2008; Brunet et al., 2009; Davidson et al., 2010; Dubois et al., 2010; Alim et al., 2011; Aufdenkampe et al., 2011; Butman & Raymond, 2011;

Neu et al., 2011; Huang et al., 2012; de Fátima F. L. Rasera et al., 2013; Salimon et al., 2013).
Studies have highlight the importance of tropical rivers to global carbon budget and the most important processes that occur in aquatic environments are the balances between photosynthesis and respiration and the gaseous exchange between water and atmosphere (Ferguson et al., 2011). Freshwater ecosystems are considered passive pipes that transport carbon from continental ecosystems to the ocean in regional and global carbon budget (Aufdenkampe et al., 2011). However,

some researchers (e.g. Richey et al., 2002; Aufdenkampe et al., 2011) have observed the amount of carbon that rivers take to ocean is only a fraction that reaches a fluvial channel from terrestrial ecosystems. Most carbon returns to the



atmosphere as carbon dioxide ($CO_2$) before it reaches the ocean, or is buried in river channels as sedimentary organic carbon (Aufdenkampe et al., 2011).

Tropical rivers largely contribute to carbon emissions. Richey et al., (2002) estimated an approximate 470TgC year[-1]

carbon evasion to the Amazon Basin, which is 13 times higher than the total organic carbon and dissolved inorganic carbon exported to the ocean by fluvial transport. Approximately 80% of all carbon are fixed in the land and transported to rivers by rainfall, whereas 20% are aquatic (Richey et al., 2002). The initial idea that rivers are passive transporters of carbon from the land to the ocean was contradicted by such values according which aquatic ecosystems are great reactors that oxidase an important part of carbon and return it to the atmosphere along the fluvial channel.

Studies that elucidate the river's role in carbon budget, regulations mechanisms, and temporal variations must be intensified. This article addresses the quantification of $CO_2$ flux in the water-air interface and the seasonal variation and its relationship with physical, chemical and biological water parameters. The work hypothesis was that the biological activity is a driving factor to CO2 exchange between air-water.

## 2 Study area

Rondônia State is located in the Western Amazon and its predominant climate is Aw - tropical wet and dry, according to the Köppen climate classification, with a defined dry period. The average annual rainfall ranges between 1400 and 2600mm/year, with rainfall indices lower than 20mm between June and August. The average air temperature varies between 24 and 26°C (Rondônia, 2012).

The Ji-Paraná River basin is located in the eastern Rondônia State, a region of highest deforestation rates in the Amazon

(Krusche et al., 2005). The Ji-Paraná River is highly impacted by agriculture and pasture between its headwaters and middle sectors (Ballester et al., 2003). It is 972km long, its width ranges between 150 and 500m and drains an area of 75400km$^2$ (Krusche et al., 2005).

The spatial distribution of Rondônia State soils is highly heterogeneous, with old and weathered soils, poor cationic characteristics, low nutrient levels, and high acidity. However, 60% of the soil of the Ji-Paraná River Basin's area are

eutrophic (predominantly latosol), rich in cations and more favorable for agriculture (Krusche et al., 2005).

## 3 Materials and methods

Field samplings were conducted on a monthly basis from September (2014) to August (2015) in an 800m stretch, in the Ji-Paraná River (Fig. 1). The $CO_2$ flux was measured according to a floating chamber methodology. The chamber (area: 0.107 m$^2$; volume 0.018207 m$^3$) was connected to a calibrated infrared gas analyzer (IRGA) (Li-COR, model LI-800) and

air was pumped to IRGA by an air pump at an approximately 160ml s$^{-1}$ flow. Each sampling campaign was replicated 10 times and the $CO_2$ flux was calculated by an empirical equation proposed by (Frankignoulle, 1988):

$$F = (\delta PCO2 / \delta t) (V/RST) \tag{1}$$

where F is the flux (molCO$_2$ m$^{-2}$ s$^{-1}$), ($\delta PCO_2 / \delta t$) is the slope of linear regression from $CO_2$ concentration in the chamber against time (valid for r$^2$ > 0.95), V is the system's volume (m$^3$), R denotes the ideal gases constant (atm m$^3$ mol$^{-1}$ k$^{-1}$), S

is the chamber's superficial area (m$^2$), and T is the air temperature (K) (Frankignoulle, 1988).

The physical-chemical parameters analyzed were pH (pH meter Orio, model 250), electric conductivity (Amber Science conductivimeter, model 2052), temperature and dissolved oxygen (oximeter YSI, model 58), turbidity (turbidimeter Quimis model TB1000), and water transparency (Secchi disc). Alkalinity was analyzed by the titrimetric method and





Cholophyll *a* was analyzed by the extraction method with hot ethanol, and the reading was performed on a spectrophotometer at 664 and 750nm wavelengths.

The air temperature was measured by an anemometer (Kestrel, model 3000) 1.5m above water and discharge and rainfall data were obtained in the HidroWeb portal of Water National Agency (Brasil, 2015).

### 3.1 Data analysis

Normality data and variance homogeneity were tested by Shapiro-Wilk and Levene tests, respectively, and showed normal distribution and homoscedasticity; therefore, parametric statistics was applied. A logarithmic conversion was applied for non-normal data. Pearson correlation matrices verified the relationship between the variables. The correlation coefficient significance was tested by *t*-Student test with 0.05 significance level and n-1 degree of freedom. The river's stages were compared by one-way ANOVA and Tukey test. All analyses were performed in R software.

### 4 Results

A defined rainfall, with periods of high and low pluviometry in summer and winter, respectively, characterizes the region (Fig. 2). Such features combined with the basin's geomorphologic characteristics influence the river dynamics. The rivers of the Amazon Basin show clear seasonality due to a defined seasonal rainfall regime and water's rise and decrease cause changes in all river dynamics. The fluviometric dynamics of the Ji-Paraná River was divided into four stages (Fig. 2), namely low water (dry stage) - July to September -, rising water (flood stage) - October to December - , high water (wet

stage) - January to March - and falling water (ebb stage) - April to June.

The discharge ranged between fluviometric periods and the highest values were achieved in the wet stage ($2448 \pm 861.46$ $m^3 s^{-1}$), whereas lower values were detected in the dry stage ($255 \pm 50.82$ $m^3 s^{-1}$). The transition periods showed intermediate values ($627 \pm 267.21$ $m^3 s^{-1}$ in the flood and $738 \pm 342.21$ $m^3 s^{-1}$ in the ebb stage). The wet stage was statically different from other fluviometric stages ($p < 0.05$). The river discharge and physical-chemical parameters are shown in

Table 1.

The water temperature ranged from 26.3°C, in December, to 29.8 C°, in September (Table 1). The dry stage showed a higher average water temperature ($27.97 \pm 1.65$ °C), whereas the ebb stage showed the lowest ($26.70 \pm 1.11$ °C). Dissolved oxygen was higher in tue dry season and lower in the wet stage ($6.31 \pm 0.29$ and $4.50 \pm 0.57$ $mgO_2$ $L^{-1}$, respectively), whereas chlorophyll was higher in the dry stage ($3.17 \pm 0.61$ mg $L^{-1}$), and lower in the wet stage, $0.87 \pm$

$0.59$ mg $L^{-1}$. It was directly correlated with dissolved oxygen ($p < 0.001$) and water transparency ($p < 0.05$) and inversely correlated with electric conductivity ($p < 0.05$) and turbidity ($p < 0.05$) (Fig. 3). Dissolved oxygen was significantly different between dry and wet water stages ($p < 0,001$), wet and flood stages ($p< 0,05$) and wet and ebb stages (0,01), and chlorophyll was different between dry and wet stages ($p <0,01$). Both electric conductivity and water transparency were lower in the dry stage ($18.25 \pm 1.05$ $\mu S$ $cm^{-1}$, $0.87 \pm 0.20$m, respectively) and higher in the wet stage ($30.42 \pm 1.52$ $\mu S$

$cm^{-1}$ $31.33 \pm 7.51$m), whereas alkalinity and pH were higher in the dry stage ($19.53 \pm 5.27$ ppm $CaCO_3$, 5.88) and lower in the ebb ($13.17 \pm 2.29$ ppm $CaCO_3$) and wet stages (5.46), respectively.

The discharge was inversely correlated to pH ($p < 0.05$), dissolved oxygen ($p < 0.001$), water transparency ($p < 0.01$), and chlorophyll ($p < 0.001$) and directly correlated to electric conductivity ($p < 0.05$) and turbidity ($p < 0.05$) (Fig. 4). The water transparency was directly related to dissolved oxygen ($p < 0.05$) and pH ($p <0.05$) and inversely related to

electric conductivity ($p < 0.01$) (Fig. 3). It was also different between dry and wet stages and dry and flood water stages ($p < 0.05$).



The Ji-Paraná River behaved as a source of carbon to the atmosphere and its emission rate ranged from $583.64 \pm 75.88$ mgC m$^{-2}$ day$^{-1}$, in July, to $5698.20 \pm 785.91$ mgC m$^{-2}$ day$^{-1}$, in February. In the dry stage, the average flux was $711.20 \pm$

$131.51$ mgC m$^{-2}$ day$^{-1}$, whereas in the wet stage, it was $4869.20 \pm 779.95$ mgC m$^{-2}$ day$^{-1}$. In the transition stages, the carbon emissions were intermediate – $2429.46 \pm 1331.67$ and $1443.79 \pm 258.57$ mgC m$^{-2}$day$^{-1}$ for to the flood and ebb stations respectively.

A statistic difference was observed between the river stages and the flux was different among wet and other water stages ($p < 0.05$). The $CO_2$ flux was directly related to electrical conductivity ($p < 0.05$) and discharge ($p < 0.001$) and inversely

related to pH ($p < 0.05$), dissolved oxygen ($p < 0.001$), water transparency ($p < 0.05$) and chlorophyll ($p < 0.001$) (Fig. 5).

## 5 Discussion

The discharge values showed a clear difference between the river stages and both water rise and decrease influenced the water physical-chemical characteristics and biological activity. The Ji-Paraná Basin's rivers show significant seasonality,

with well-defined wet and dry periods, which result in substantial changes in the water level and surface area (Rasera et al., 2008).

Fewer oxygen concentrations were detected in the wet season, which evinces an increase in the water column respiration caused by the entrance of organic matter by runoff and overflow of water for floodplain. Various forms, fractions, and decomposition states of organic matter reach rivers from their watersheds and their composition and concentration are

modified by metabolic processes along the fluvial channel. Therefore, part of the carbon may be outgassed to the atmosphere as $CO_2$ (Rasera et al., 2008). In the dry stage, the water turbulence increases due to the formation of turbulence areas and causes an oxygen dissolution. The relationship between dissolved oxygen and discharge (Fig. 4) strengthens this contestation.

The high correlation between chlorophyll and dissolved oxygen indicates the biologic activity strongly influences the

oxygen variation and carbon cycle dynamics through photosynthesis. The larger oxygen concentration was related to the highest values of chlorophyll and water transparency and the smaller concentration of chlorophyll was associated with the lowest oxygen values. Such results indicate a biological control of aquatic oxygen.

Water transparency is a very important parameter for aquatic ecosystems, once it characterizes the depth of the illuminated region and can provide information about increase or decrease of dissolved organic matter and suspense particulate matter

(Wetzel and Likens, 2000). The relationship between water transparency and chlorophyll (Fig. 3) indicates light penetration is a regulating factor in the primary production for the Ji-Paraná river. The decrease in the water transparency in the wet season suggests an increase in the dissolved and particulate matter from terrestrial areas.

The Ji-Paraná River showed slightly acid water, with lower values occurring in the wet season. Salimon et al., (2013) detected low pH values (7.2) and big $pCO_2$ ($4029 \pm 551$ppm) in the wet season in Purus River, whereas pH was higher

(8.2) and $pCO_2$ lower ($780 \pm 152$ppm) in the dry season. The inverse and moderate relationship between pH and flux (Fig. 5) suggests the same behavior was observed in the Ji-Paraná River.

The $CO_2$ flux dynamics follows the flooding pulse of the river, with higher values in the wet stage and lower ones in the dry stage, and a 6.8 ratio between the stages. Several studies conducted in the Amazon have detected the same pattern, with differences in the carbon evasion between the periods (wet and dry stages) ranging from 1.1 to 44 times (Gomes,

2009; Neu et al., 2011; Rasera et al., 2008, 2013; Richey et al., 2002; Salimon et al., 2013; Sousa, 2013).



The strong correlation between flux and chlorophyll concentration (Fig. 5) indicates the primary production regulates the $CO_2$ flux. Studying rivers in the Amazon Basin, Rasera et al., (2013) found negative flux values in Araguaia, Javaés and Teles Pires rivers in the dry stage and associated this behavior with the water transparency that favors light penetration and primary production. In all rivers, $pCO_2$ was strongly and inversely correlated to dissolved oxygen and pH and directly

associated with dissolved organic carbon, which evidences a significant relationship between $pCO_2$ and aquatic metabolism. Sand-Jensen and Staehr, (2012) associated higher flux and $CO_2$ concentration in water with lower chlorophyll values in two streams in Denmark. Dawson et al., (2004) found undersatured $pCO_2$ conditions in Scotland streams and attributed them to high photosynthetic activity. Our results suggest the biologic activity influences $CO_2$ air-water exchanges in the Ji-Paraná river.

On the other hand, the $CO_2$ emission increases in the wet stage. The higher precipitation leads to the input of $CO_2$ and allochthonous organic matter from soil, riparian zone, and wetland, and carbon, which would be evaded directly from terrestrial systems, is outgassed from aquatic ecosystems (Butman and Raymond, 2011). Yet, the photosynthetic activity in Amazon rivers leads to high respiration rates through on a labile substrate for heterotrophs (Ellis et al., 2012), which helps the explanation of the positive flux values in the dry season, when the carbon inputs from terrestrial ecosystems are

reduced. Therefore, autochthon carbon sustains the positive values of flux.

According to Richey et al., (1988), the aquatic environments of the Amazon Basin are characterized by positive values of free $CO_2$ and Apparent Oxygen Utilization - AOU (excess of respiration over photosynthesis) - and a general association between high levels of AOU and high free $CO_2$. In our study, the strong and inverse relationship between flux and dissolved oxygen and chlorophyll strengthens those results.

All periods showed positive flux values and several studies have found the same tendency (Alin et al., 2011; Aufdenkampe et al., 2011; Borges et al., 2004; Butman and Raymond, 2011; Cole and Caraco, 2001; Davidson et al., 2010; Dawson et al., 2004; Dubois et al., 2010; Gupta et al., 2008; Ho et al., 2007; Johnson et al., 2008; Jones et al., 2003; Neu et al., 2011; Prasad et al., 2013; Rasera et al., 2013; Richey et al., 2002; Salimon et al., 2013; Sand-Jensen and Staehr, 2012). In low-order systems, the higher $pCO_2$ concentration and $CO_2$ flux are sustained by groundwater and high-order rivers and

estuaries show a lower air-water concentration gradient, but a larger contribution of carbon emission to the atmosphere due to bigger superficial areas.

## 6 Conclusions

The Ji-Paraná River behaves as a source of $CO_2$ to the atmosphere and follows the Amazon Basin Rivers tendency, showing a significant temporal variability with higher flux values in the wet stage and lower values in the dry one.

Dissolved oxygen was the chemical parameter that better explained such a flux variability. The strong relationship between chlorophyll and $CO_2$ flux is responsible for the lowest $CO_2$ emission rates in the dry seasons, when the water transparency decreases and the primary production increases. The autochthone respiration can explain the positive flux values in the dry season and evidences the excess of respiration over photosynthesis in all hydrologic periods. Our results corroborate to the understanding of the role of tropical rivers in the global carbon budget and suggest biological activity

is an important factor for the regulation of the aquatic carbon dynamics.



## 7 Acknowledgments

The authors acknowledge the National Council of Scientific and Technology Development (Conselho Nacional de Desenvolvimento Científico e Tecnológico - CNPq) for the funding and Ji-Paraná Fire Department for the logistical support.

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




**Table 1: CO₂ flux, discharge, and physical-chemical water parameters**

| Month/ year | CO₂ Flux (mgCm⁻² day⁻¹) | T¹ (°C) | DO² (mgL⁻¹) | EC³ (μS cm⁻¹) | pH | Alkalinity (ppm CaCo₃) | WT⁴ (m) | Turbidity (NTU) | Chlorophyll. (mg L⁻¹) | Discharge (m³ s⁻¹) |
|---|---|---|---|---|---|---|---|---|---|---|
| Sep/14* | 846,33 ± 209,98 | 29,80 | 6,65 | 19,45 | 5,53 | 22,00 | 0,66 | 16,03 | 3,85 | 214 |
| Oct/14** | 999,18 ± 170,57 | 29,50 | 5,94 | 11,10 | 5,82 | 9,17 | 0,53 | 24,60 | 2,99 | 387 |
| Nov/14** | 2655,68 ± 511,27 | 26,92 | 5,28 | 44,00 | 5,74 | 16,50 | 0,16 | 124,00 | 1,71 | 915 |
| Dec/14** | 3633,53 ± 386,13 | 26,30 | 5,63 | 27,00 | 5,82 | 16,13 | 0,28 | 57,13 | 1,28 | 578 |
| Jan/15*** | 4149,93 ± 388,18 | 29,72 | 5,11 | 28,70 | 5,91 | 19,07 | 0,27 | 46,67 | 1,54 | 1600 |
| Feb/15*** | 5698,20 ± 785,91 | 26,70 | 3,98 | 31,00 | 5,26 | 14,30 | 0,40 | 33,63 | 0,43 | 3323 |
| Mar/15*** | 4759,47 ± 359,75 | 26,90 | 4,42 | 31,56 | 5,44 | 20,17 | 0,27 | 47,53 | 0,64 | 2421 |
| Apr/15**** | 1547,68 ± 210,58 | 27,70 | 5,86 | 29,50 | 6,09 | 15,81 | 0,32 | 49,77 | 1,60 | 1071 |
| May/15**** | 1634,26 ± 252,75 | 26,90 | 5,88 | 25,20 | 5,52 | 11,95 | 0,49 | 49,57 | 1,28 | 756 |
| Jun/15**** | 1149,43 ± 230,59 | 25,50 | 6,27 | 26,53 | 6,15 | 11,73 | 0,63 | 16,23 | 2,20 | 387 |
| Jul/15* | 583,64 ± 75,88 | 26,60 | 6,11 | 17,50 | 6,14 | 13,48 | 0,89 | 14,50 | 2,67 | 312 |
| Aug/15* | 703,62 ± 19,48 | 27,50 | 6,18 | 17,80 | 6,62 | 23,10 | 1,06 | 10,02 | 2,99 | 238 |

¹Water Temperature, ²Dissolved Oxygen, ³Electric Conductivity, ⁴Water transparency.

*Low Water (dry stage), **Rising Water (flood stage), ***High Water (wet stage), ****Falling Water (ebb stage).





**Study Area in the Ji-Paraná River**

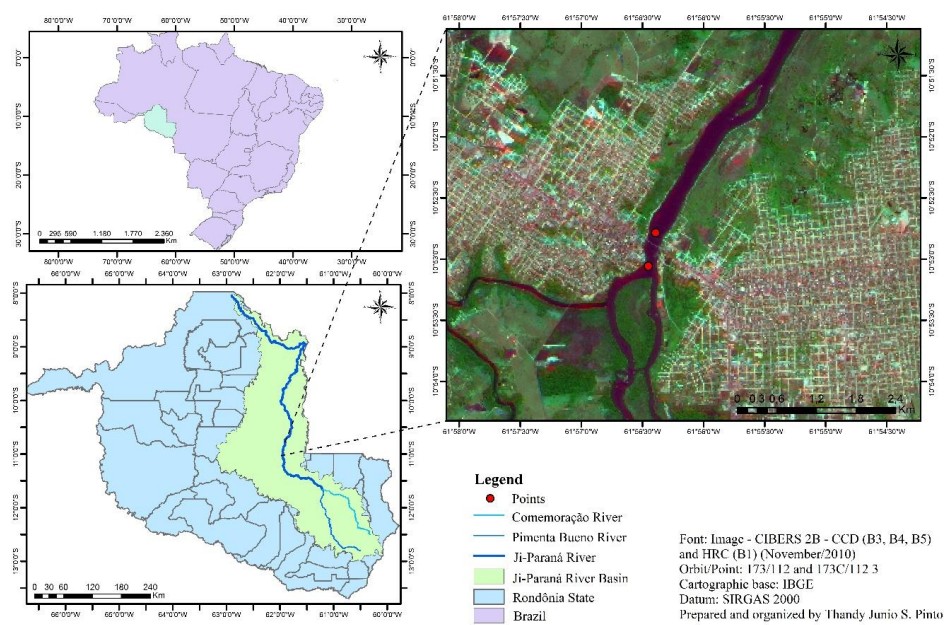


**Figure 1: Map of the Ji-Paraná River basin, samplings were conducted between points in an 800m stretch.**









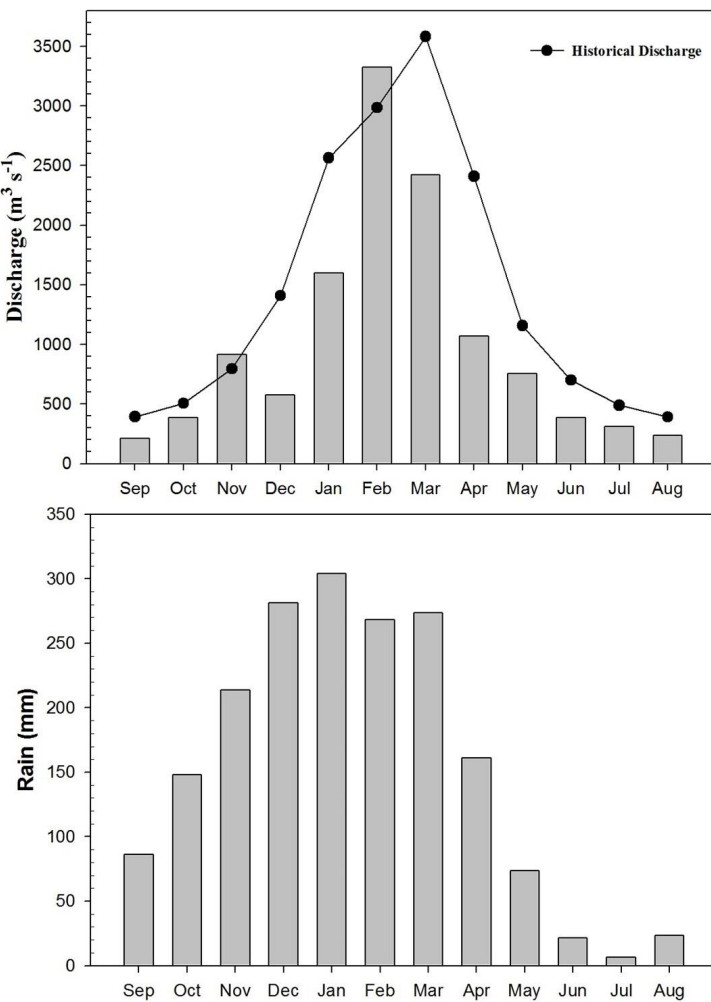

**Figure 2: Recent historical average discharge (1996-2006) and discharge between September (2014) and August (2015) for the study period (2014-2015) and historical average rainfall (1976-1996) for the Ji-Paraná River in the sample area.**

Font. – Data from National Agency of Water (Brasil, 2015).






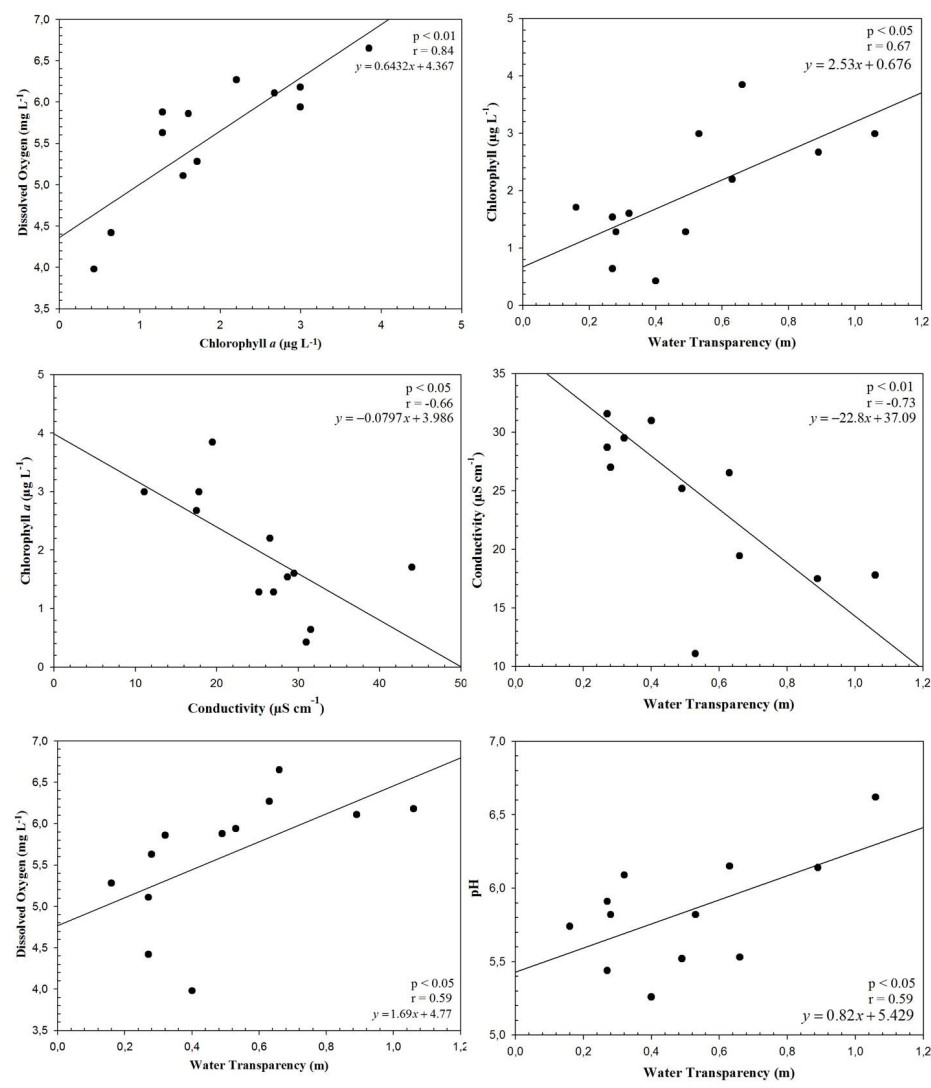

**Figure 3: Relationship among physical, chemical and biological variables.**



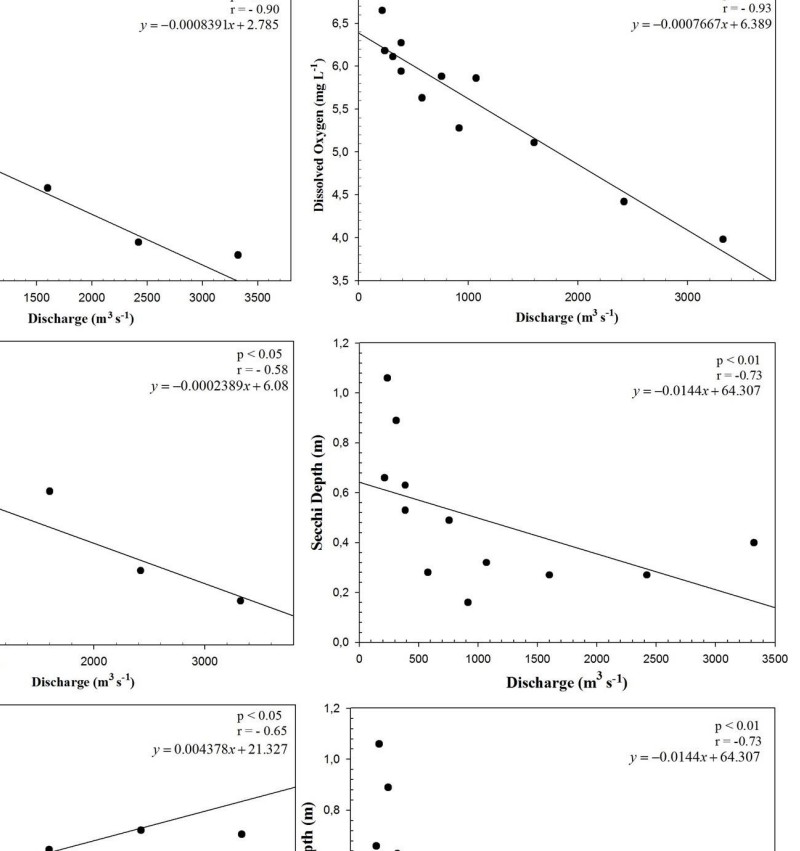

**Figure 4: Relationship among discharge and physical-chemical parameters.**





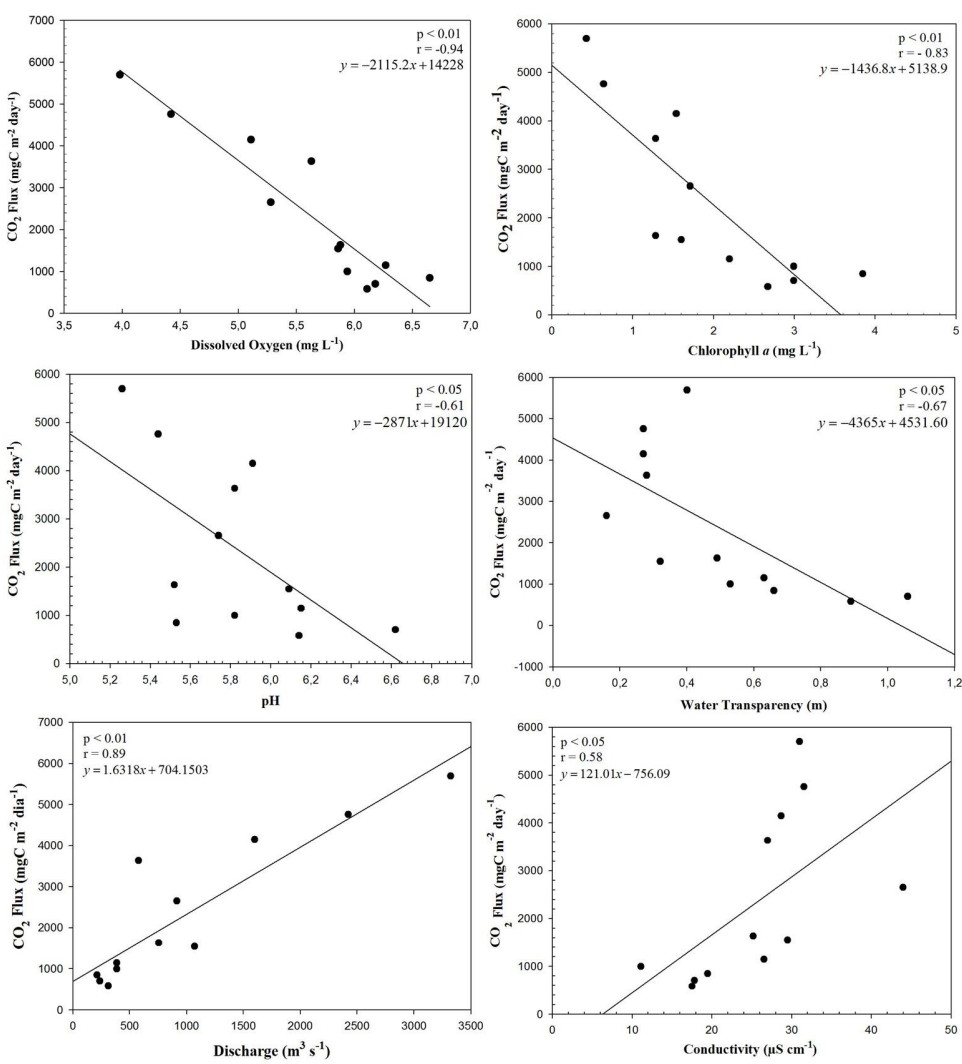

**Figure 5: Relationship among physical-chemical parameters and CO₂ flux.**
