# Peer review of "CO2 Flux and its Relationship with Water Parameters and Biological Activity in the Ji-Paraná River (Rondônia State – Western Amazon)"

_Biogeosciences, 2017_

## Referee Comment (RC1) · Anonymous Referee #1 · 16 Nov 2017

1. Does the paper address relevant scientific questions within the scope of BG? The Ms. covers a topic of high current scientific interest, well within the scope of BG. However, the Ms. in its present form does not address a scientific question on a level high enough for publication in BG (see further comments)

2. Does the paper present novel concepts, ideas, tools, or data? The Ms. does not present any novel concepts, ideas or tools. In fact, it feels like the authors would befit from reading some of the latest work published in the subject (e.g. Hotchkiss et al. 2015, Stanley et al. Rasilo et al 2015, Butman et al. 2016, Lauerwald et al. 2015,

[Figure]

Lundin et al. 2014, Bastviken et al 2010) to develop the Ms. ecological/biogeochemical concept/discussion. The data is not unique as that similar/equal data from other research groups have been published before. Anyhow, the authors have put considerable efforts into sampling and analyses and should be acknowledged for the good dataset presented in this Ms., because for the scientific development of the current field it is very important that good original data (from the whole globe) is published and shared. Therefore, I'm not only hoping but also strongly encouraging the authors to resubmit the Ms. after significant revisions.

3. Are substantial conclusions reached? As it is written now, considering the potential of the data set, the conclusions are not of satisfaction. The authors draw the conclusion that biological activity is an important factor for the regulation of aquatic carbon. This is not any new knowledge, in fact, this is basic knowledge expected to be known by the target reader. It could be implicitly be understood that the authors means photosynthesis when they write biological activity, which is just not correct.

4. Are the scientific methods and assumptions valid and clearly outlined? The Materials and methods section should better describe the methods used and/or preferably be properly cited (see specific comments). More emphasis should be put on describing the flux estimations.

5. Are the results sufficient to support the interpretations and conclusions? The discussion and conclusion part need sufficient rewriting (see specific comments)

6. Is the description of experiments and calculations sufficiently complete and precise to allow their reproduction by fellow scientists (traceability of results)? The authors present all data in a Table 1, which allow others to use the data and validate the statistics. However, the description of the field methods is unclear. Were the chambers traveling or were they static? If floating down stream, what was the traveling time? What time of the day was sampling performed? Was sampling always performed at the same hour? How was water sampled and on what depth? What depth was used for

the in-situ measurements?

7. Do the authors give proper credit to related work and clearly indicate their own new/original contribution? The last couple of years there have been extreme developments in the field of stream/river biogeochemistry. I think the authors would benefit from reading some of the recent work (see previous comments)

8. Does the title clearly reflect the contents of the paper? I don't think the authors explain/discuss enough what they mean with biological activity in the paper, which makes the title a bit misleading

9. Does the abstract provide a concise and complete summary? The abstract overall reflects the Ms. However, as significant rewriting is required; it is natural that the abstract is changed accordingly

10. Is the overall presentation well structured and clear? The Ms. is overall well-structured and written short and concise.

11. Is the language fluent and precise? The Ms. needs a linguistic (professional/native speaker) review

12. Are mathematical formulae, symbols, abbreviations, and units correctly defined and used? Overall, it looks good but significant figures (decimals) needs to be adjusted throughout the Ms. There are too many examples of overestimated precisions (e.g. pH, $CO_2$ flux, etc). Further, the precision of SD can never be better than the precision of the estimated parameter (see L92)

13. Should any parts of the paper (text, formulae, figures, tables) be clarified, reduced, combined, or eliminated? I suggest the authors to condense the figures 3-5. I also suggest that Table 1 is moved to the supplementary material in favor for a figure showing the most important data over time.

14. Are the number and quality of references appropriate? The authors use 22 references in Ms. of which 13 refers to the second sentence. For comparison, a short

format journal often limits the references to 30. Considering my earlier criticism on referring to recent literature, I believe it must be room for some more references.

15. Is the amount and quality of supplementary material appropriate? I suggest a supplementary description of the flux measurements.

Specific Comments (major and minor) Abstract The whole Ms needs substantial revision. The abstract needs to be adjusted accordingly. L17 (and throughout the whole Ms.): Adjust the number of significant figures.

Introduction

L28: Limit the numbers of references.

L31: This sentence implies that tropical rivers are of special interest and that this is the consensus. Why? Pleas provide references. Is it fair to state that the balance between photosynthesis and respiration together with water-atm exchange is the over-all most important processes in rivers? What is this statement based on, only Ferguson et al. 2011?

L33: This sentence is just not correct. The research community have known for almost 20 years that freshwater systems are active pipes (starting with Cole et al. 1994) and the latest IPCC reports do acknowledge that. Please rephrase the sentence.

L39: The sentence is not clear. I think the authors mean "from the Amazon Basin" not " to the Amazon Basin".

L41: The sentence is not clear. Pleas rephrase.

L42: Rephrase the sentence

L45: Please motivate why more studies are needed and/or add references.

L46: I don't agree that this Ms. contains a quantification of $CO_2$ emissions. The authors estimated vertical $CO_2$ fluxes and relate them to the biogeochemical properties of the

river.

L48: What biological activity? It is obvious that CO2 concentrations are related to some kind of biological activity but where and what kind?

Material and Methods

L62: Please also provide the hours of the measurements, e.g. mid-day, evening, morning etc.

L63. Provide more information on how the flux measurements were preformed and/ or add a reference. For example, was the chamber static or floating down stream? How long was the sampling time? I suggest a longer description in the supplementary materials.

L64: Is a linear equation a good choice of a model for this purpose? My concern is that the accumulation of CO2 in the chamber is not linear (see Bastviken et al. 2010), resulting in underestimations of CO2 fluxes, when traveling times are long. Anyhow the choice of model/calculation should be motivated in the methods/sup. material.

L71 and throughout the whole section: What depth for water sampling and measurements?

L73 and throughout the Ms.: There is more than one method for titration, change accordingly.

L73: How long before the water was analyzed and how was the water stored until then?

L76: Temperature is not measured by an anemometer. Change "rainfall" to precipitation'

Data analysis

L82: it is not clear what the ANOVA tests. Is it between years or is it within years. If within years, is this test needed?

Results

L89: Is it not enough to have only one name for each stage?

L91: A bit obvious sentence. The dry stage is called dry because water levels are low.

L94: see previous comment. Is this ANOVA really needed?

L100 – 104: Maybe present the Pearson correlation matrix as a Table instead?

Discussion Overall, I think the discussion needs significant revisions, as it in its present state does not read very well. Some parts are very hard to follow. The authors need to better build up a 'reaction chain' explaining drivers/controls, starting with the discharge variation and how that relates to the biogeochemistry. As it is written now it feels like the authors don't have a good understanding of the causal relationships between properties. Sometimes the authors even preset "circular" chain of evidence or/and contradicting arguments. In addition, the authors make some crucial assumptions which are neither well argued for nor properly cited. For example, it is assumed that organic C concentrations increase during high flood, resulting in increased respiration and $CO_2$ efflux. However, this statement is not supported by original data or properly cited literature. Further, what role play nutrients (N, P) in this story? Are nutrients assumed not to be of any importance? Maybe they are not, but that needs to be tested before being assumed.

L127: Although it is possible that the decrease in $O_2$ concentration is caused by an increase in respiration due to increased inputs of terrestrial organic C, it is a bit simplistic to just conclude this in one sentence. I think some further discussion is needed. Are river organic C concentrations for example higher during this period? In addition, I can't help noticing that Chl. a is low during the same period. What is the connection there?

L128 – 131: To me it is not clear what the authors want with these sentences.

L160 – 163: This sentence does not make any sense.

L163 - 165: I think the authors need to explain this better. A long-term overpressure of $CO_2$ (relative atm) requires external inputs of carbon, i.e autochthonous C can therefore not the only responsible for long-term $CO_2$ saturation. Anyhow, shorter periods of $CO_2$ saturation fed by autochthonous C is theoretically possible but that requires a storage of aquatic produced organic C.

166: This argumentation is contradictory to the previous discussion (L163 -165).

L168 – 169: I don't agree that the inverse relationship between $CO_2$ (?) flux and O2 simply can be interpreted as that respiration exceeds photosynthesis.

L174: pCO2 is the partial pressure of $CO_2$, not concentration. These two last sentences do not fit in here.

Conclusions The discussion needs significant revisions. Change the conclusion accordingly

References

Bastviken et al. 2010, Methane emissions from Pantanal, South America, during the low water season: toward more comprehensive sampling

Butman et al. 2016, Aquatic carbon cycling in the conterminous United States and implications for terrestrial carbon accounting

Cole et al. 1994, Carbon dioxide supersaturation in the surface waters of lakes

Hotchkiss et al. 2015, Sources of and processes controlling $CO_2$ emissions change with the size of streams and rivers

Lauerwald et al. 2015, Spatial patterns in $CO_2$ evasion from the global river network

Lundin et al. 2014, Integrating carbon emissions from lakes and streams in a subarctic catchment

Rasilo et al 2015, Large‐scale patterns in summer diffusive CH4 fluxes across boreal lakes, and contribution to diffusive C emissions

Stanley et al. 2016, The ecology of methane in streams and rivers: patterns, controls, and global significance

---

## Referee Comment (RC2) · Anonymous Referee #2 · 14 Dec 2017

Dear authors,

This study reports the CO2 flux and other parameters in the Ji-Paraná River basin. The result is organized in a table and three figures. However, the details of measurements are unclear. The statistic method can be largely improved. The discussion and conclusion did not provide quantified number (not just statistic numbers) to bring a new insight.

Major comments:

1. The authors use correlations among the parameters in Table 1. There is at least one better way to deal with the correlations, for example, PCA (Principle component analysis). The PCA can be easily done through free software "R". The explanation for the result from PCA should be based on the beigeochemical knowledge. There are several high correlations in Fig. 3 to 5. The authors can consider to discuss them systematically instead of individually. In addition to statistics, Discussion should give readers a better insight for Biogeoscience. How strong is biological activity? Primary production and respiration are mentioned but are not quantified. Are these number high or low by comparing to other systems? What is the role of river discharge in addition to biological activity? The correlation between them was still significant.

2. The discussion of AOU, pCO2 is questionable. The oxygen in river water should be able to exchange with the atmosphere. The temperature of river water can also varied day and night. How theses effects can change DO concentration in the water? Line 174, "pCO2" is the partial pressure of carbon dioxide, not concentration.

3. The writing is sloppy. The structure should be modified. Section 2 and 3 should be combined. Since the authors only have 3.1 and no 3.2. Please consider to use a better structure to organize them. Details in methods are lacking. Precision for the methods, such as pH, alkalinity, and others, should be provided. How the uncertainty of F is calculated? Line 170- 174, there are 18 references in one sentence. Please select references and merely use them.

4. The conclusion did not provide a new insight. Such a conceptual conclusion has well known.

Minor comments: Line 50 to 60, there are 3 paragraphs in 10 lines. Please modify the structure.

Line 126, "Fewer oxygen concentrations"? Do you mean "lower"?

Line 189, "Acknowledgments", please check spelling.
All the "2" in CO2 in the references should be modified to LOWER CASE.

Line 232, "exchange1"?

Table 1, the authors use commas for all numbers? I think the authors want to use "."
* * *